# *Toxoplasma gondii* and *Neospora caninum* Infections in Stray Cats and Dogs in the Qinghai–Tibetan Plateau Area, China

**DOI:** 10.3390/ani12111390

**Published:** 2022-05-28

**Authors:** Jinfang Yang, Jingkai Ai, Tongsheng Qi, Xiaomin Ni, Zichun Xu, Liangting Guo, Yali Sun, Ying Li, Ming Kang, Jixu Li

**Affiliations:** 1State Key Laboratory of Plateau Ecology and Agriculture, Qinghai University, Xining 810016, China; yangjinfang@qhu.edu.cn (J.Y.); 1810107052@qhu.edu.cn (J.A.); qitongsheng@qhu.edu.cn (T.Q.); 2013990004@qhu.edu.cn (X.N.); xzc3045765902@hotmail.com (Z.X.); jy702667594@hotmail.com (L.G.); yalisun@qhu.edu.cn (Y.S.); 2000990008@qhu.edu.cn (Y.L.); 2013990003@qhu.edu.cn (M.K.); 2College of Agriculture and Animal Husbandry, Qinghai University, Xining 810016, China; 3Qinghai Provincial Key Laboratory of Pathogen Diagnosis for Animal Diseases and Green Technical Research for Prevention and Control, Qinghai University, Xining 810016, China

**Keywords:** *Toxoplasma gondii*, *Neospora caninum*, stray cat, stray dog, Qinghai–Tibetan Plateau Area

## Abstract

**Simple Summary:**

Diseases caused by parasites have introduced serious threats to human health and the development of animal husbandry in the Qinghai–Tibetan Plateau Area (QTPA), such as toxoplasmosis caused by *Toxoplasma gondii* and neosporosis caused by *Neospora caninum*. However, information on the epidemiology of toxoplasmosis and neosporosis in stray cats and dogs which are the definitive hosts of *T. gondii* and *N. caninum* in the QTPA is limited. The aim of this study was to establish a detailed record of the seroprevalence of *T. gondii* and *N. caninum*-specific IgG and IgM antibodies in serum samples and the molecular epidemiology in feces from stray cats and dogs in the plateau area. The results revealed that stray cats and dogs in the QTPA, China present both *T. gondii* and *N. caninum* infection through the antibodies and antigen detection of the indirect ELISA tests and qPCRs. The present study suggests the prevalence of acute neosporosis and chronic re-emergence of toxoplasmosis in stray cats and dogs in the testing area. To the best of our knowledge, this is the first report of *T. gondii* and *N. caninum* infection in cats and dogs in the QTPA, and the first determination of *N. caninum* infection in cats in China. In conclusion, stray cats and dogs play key roles in the transmission and prevalence of *T. gondii* and *N. caninum* in the plateau area.

**Abstract:**

*Toxoplasma gondii* and *Neospora caninum* belong to the Apicomplexan protozoa which is an obligate intracellular parasite, causing toxoplasmosis and neosporosis throughout the world. Cats and dogs are the definitive hosts of these two parasites. However, information on the epidemiology of toxoplasmosis and neosporosis in stray cats and dogs in the Qinghai–Tibetan Plateau Area (QTPA) is limited, and little is known about the diversity of the diseases. Therefore, the aim of this study was to perform indirect ELISA tests based on recombinant *Tg*SAG1, *Tg*GRA1, *Nc*SAG1 and *Nc*GRA7 proteins to establish a detailed record of the seroprevalence of *T. gondii* and *N. caninum*-specific IgG and IgM antibodies in serum samples and to develop qPCR amplification based on *Tg*B1 and *Nc*Nc5 genes to conduct molecular epidemiology in feces from stray cats and dogs in the QTPA. In the current study, a total of 128 cat serum samples were analyzed through serological tests in which 53 (41.4%) and 57 (44.5%) samples were found positive for *T. gondii* specific-IgG and IgM antibodies, and 2 (1.6%) and 74 (57.8%) samples were confirmed positive for *N. caninum* specific-IgG and IgM antibodies, respectively. Out of 224 stray dog sera, 59.8% and 58.9% were recorded as positive against anti-*Toxoplasma* IgG and IgM antibodies, 17.9% and 64.7% were detected positive against *Neospora* IgG and IgM. On the other hand, 1 of 18 cat fecal samples was successfully amplified within the Ct value of 10 to 30 while no cat was positive for neosporosis. Moreover, a higher prevalence of toxoplasmosis in stray dogs (14.5%, 16/110) than of neosporosis (5.5%, 6/110) with different parasite numbers were found. Further analysis showed that no significant sex differences were found nor between the overall infection rates of *T. gondii* and *N. caninum* in this study. This study suggests that stray cats and dogs play key roles in the transmission and prevalence of *T. gondii* and *N. caninum* in the plateau area.

## 1. Introduction

The Qinghai–Tibetan Plateau Area (QTPA), the largest plateau with the highest average altitude on the planet (an average elevation of more than 4000 m above sea level), includes Tibet, Qinghai, western Sichuan, and southern Xinjiang (the total area is about 2.5 million square kilometers) [1]. Qinghai province in China is an important part of the QTPA [2,3,4]. QTPA has abundant pasture resources and a variety of livestock. Therefore, animal husbandry is an important industry in the province. However, diseases caused by parasites have introduced serious threats to human health and the development of animal husbandry.

*Toxoplasma gondii* is a zoonotic pathogen that can infect almost all warm-blooded animals including humans, causing toxoplasmosis [5,6,7,8,9]. Cats and other members of the family *Felidae* serve as the definitive hosts of *T. gondii* [10]. Oocysts shed by these definitive hosts cause wide-ranging environmental contamination [10]. The intermediate hosts living in the QTPA such as yaks, Tibetan sheep, and horses could be infected by the ingestion of food or water contaminated with oocysts. *Neospora caninum* is a protozoan parasite that causes neosporosis resulting in abortion in cattle as well as reproduction problems and neurological disorders in dogs [11,12,13,14]. Dogs are the definitive hosts shedding oocysts in the environment that play an important role in the epidemiology of *N. caninum* [15,16]. Dogs are a major risk factor for the occurrence of miscarriages and stillbirths associated with *N. caninum* in cattle and other intermediate hosts [15,16].

Although several studies have examined the seroprevalence of *T. gondii* in animals in the QTPA [17,18,19,20,21], to date, the information about the epidemiology of toxoplasmosis and neosporosis in stray cats and dogs in the QTPA is limited, and little is known about the diversity of the diseases. The surface antigen 1 (SAG1) and dense granule protein 7 (GRA7) of *T. gondii* and *N. caninum* have been identified and tested as important candidates for the serological diagnosis of toxoplasmosis and neosporosis [22,23,24,25,26], while the *Tg*B1 and *Nc*Nc5 genes have been used to molecularly amplify the *T. gondii* and *N. caninum* in quantitative PCR (qPCR) from fecal and tissue DNA [27,28,29,30,31]. Therefore, the aim of this study was to perform indirect ELISA tests based on recombinant *Tg*SAG1, *Tg*GRA1, *Nc*SAG1 and *Nc*GRA7 proteins and qPCR amplification based on the *Tg*B1 and *Nc*Nc5 genes to establish a detailed record of the seroprevalence of *T. gondii* and *N. caninum*-specific IgG and IgM antibodies in serum samples and the molecular epidemiology in feces from stray cats and dogs in the QTPA.

## 2. Materials and Methods

### 2.1. Sample Collection

In this study, a total of 352 blood samples were collected from all stray cats and dogs in a shelter in Qinghai Province on the QTPA, China, with the geographical coordinates of 36°34′ N and 101°49′ E from September 2021 to February 2022, to evaluate the seroprevalence and epidemiology of *T. gondii* and *N. caninum* in two species, including stray cats (*n* = 128) and dogs (*n* = 224). Feces from 128 stray animals, 18 from stray cats and 110 from stray dogs, were collected from shelter to amplify *T. gondii* and *N. caninum* parasites in the fecal DNA.

We selected the cephalic vein of dogs and cats for blood collection; these blood samples were kept at 37 °C for 30 min, centrifuged at 5000 rpm for 10 min at 4 °C to separate and harvest sera and stored at −20 °C until used. All collected feces were fresh. All feces of cats were collected from the rectum, and vast majority of feces of dogs were collected from rectum, and a small portion was collected from soil. Due to the aggressive nature of some stray dogs, we cannot collect feces from rectum, but can only collect samples from the center of the feces immediately after they defecate. Only some animals in the shelter had their own individual cages; so, we only collected feces from these animals to rule out the possibility that feces from the same animal could be collected twice. All procedures were carried out according to the ethical guidelines of Qinghai University.

### 2.2. Indirect ELISA

The recombinant r*Tg*SAG1, r*Tg*GRA1, r*Nc*SAG1 and r*Nc*GRA7 were expressed using the previously described methods [22,23,24,25,26], with slight modifications. The protein concentration was measured with a bicinchoninic acid protein assay kit (Thermo Fisher Scientific, Inc., Rockford, IL, USA). The 1 μg/mL recombinant proteins were diluted in coating buffer (0.05 M Carbonate-Bicarbonate, pH 9.6) to perform indirect ELISA analysis: the current sera were diluted by 1:100 and the HRP conjugated anti-cat and dog IgG or IgM secondary antibodies (Abcam, Cambridge, England(Goat-anti-Cat IgG: ab112801, Goat-anti-Cat IgM: ab112792, Goat-anti-Dog IgG: ab112852, and Goat-anti-Dog IgM: ab112835)), were diluted 1:4000. In this study, ABTS (2,2′-azino-bis(3-ethylbenzothiazoline-6-sulfonic acid)) substrate was used to show the results at OD 415 nm. For the resulting judgment, the mean X and standard deviation SD of the negative control results were calculated, and X+3SD was cutoff value of the GRA7-ELISA and SAG1-ELISA. The OD 415 values were higher than the respective cutoff values, and when the OD 415 values of the same sample for both rGRA7 and rSAG1 detection/the negative control OD 415 values were both greater than or equal to 2.1, the result was considered positive. The positive and negative serum samples of mice for *T. gondii* (kept at our lab) or the positive serum samples of mice for *N. caninum* (gift from Prof. Lijun Jia from Yanbian University, Jilin, China) were set as the controls to confirm the indirect ELISAs.

### 2.3. DNA Isolation and Quantitative PCR (qPCR) Detection

DNA was extracted from the feces of stray cats and dogs by Fecal DNA extraction Kit (Tiangen, China), according to the manufacturer’s instructions. The 50 ng fecal DNA was then amplified with primers specific to the *T. gondii* B1 gene (forward primer: 5′-AAC GGG CGA GTA GCA CCT GAG GAG-3′ and reverse primer: 5′-TGG GTC TAC GTC GAT GGC ATG ACA AC-3′) and *N. caninum* Nc5 gene (forward primer: 5′-ACT GGA GGC ACG CTG AAC AC-3′ and reverse primer: 5′-AAC AAT GCT TCG CAA GAG GAA-3′) by qPCR. The amplification was performed with 50 ng DNA, 5 μL PowerUp SYBR Green Master Mix (Thermo Fisher Scientific, Inc., Waltham, MA, USA), and 500 nM concentrations of gene-specific primers in a 10 μL total reaction volume using the standard protocol recommended by the manufacturer (2 min at 50 °C, 10 min at 95 °C, 40 cycles of 95 °C for 15 s, and 60 °C for 1 min). Amplification, data acquisition, and data analysis were carried out using the ABI Prism 7900HT sequence detection system (Applied Biosystems, Foster, CA, USA), and the cycle threshold values (Ct) were calculated. When the Ct value was between 10 and 30, we considered the result to be valid and proceeded to the calculation of the number of parasites. A standard curve was constructed using 10-fold serial dilutions of *T. gondii* and *N. caninum* DNA extracted from 10^5^ parasites; thus, the curve ranged from 0.01 to 10,000 parasites. The parasite number was calculated from the standard curve as described above [27,28,29,30,31].

### 2.4. Statistical Analysis

To graph and analyze the data, GraphPad Prism 8 software (GraphPad Software Inc., San Diego, CA, USA) was used. The prevalence and 95% confidence intervals per pathogen species were calculated using the OpenEpi program (http://www.openepi.com/Proportion/Proportion.htm, (accessed on 4 April 2022)). The chi-square test was used to compare proportions of detected sample positivity in different regions and among different animals. The differences were considered to be statistically significant when the resulting *p*-values were lower than 0.05.

## 3. Results

Both r*Tg*/*Nc*SAG1 and GRA7-based indirect ELISA in this study were used to detect the *T. gondii-* and *N. caninum-*specific IgG and IgM antibodies in 352 stray cats and dogs in the QTPA (Figure 1). There was no cross-reaction between the recombinant proteins of *T. gondii* and *N. caninum* and another parasitic antibody in the detection, and they were each unique (Figure 2). In the current study, 128 cat serum samples were analyzed through serological tests in which 53 (41.4%) and 57 (44.5%) samples were found positive for *T. gondii-*specific IgG and IgM antibodies, and 2 (1.6%) and 74 (57.8%) samples were confirmed positive for *N. caninum-*specific IgG and IgM antibodies, respectively (Table 1). Out of 224 stray dog sera, 59.8% and 58.9% were recorded as positive against anti-*Toxoplasma* IgG and IgM antibodies, and 17.9% and 64.7% were detected positive against *Neospora* IgG and IgM (Table 1).

In this study, 1 of 18 cat fecal samples was successfully amplified, with a Ct value between 10 and 30 through the qPCR-based *Tg*B1gene (Figure 3A), while no DNA was positive for neosporosis (Table 2). The present study recorded a higher prevalence of toxoplasmosis in stray dogs (14.5%, 16/110) than of neosporosis (5.5%, 6/110) with different parasite numbers in 50 ng fecal DNA (Table 2 and Figure 3B). Three indicators, IgG, IgM and qPCR, were used to analyze toxoplasmosis and neosporosis in 352 sera and 128 non-corresponding feces in which 248 (51.7%) samples were positive for at least one indicator, and 38 (7.9%), 111 (23.1%), and 1 (0.2%) samples were coinfected with *T. gondii* and *N. caninum* using IgG, IgM, and qPCR detection, respectively (Table 3).

The sex-wise prevalence of *T. gondii* and *N. caninum* was higher in female stray cats (41.6% and 29.8% as compared to the male population of 39.2% and 25%, respectively), with no significant difference (Table 4). The infection rate (40.4%) of neosporosis in female stray dogs was significantly higher than the 27.2% prevalence in males (*p*-value = 0.02), but toxoplasmosis was consistent (Table 4). Comparing the prevalence of toxoplasmosis or neosporosis between the tested cat and dog samples in the current area, it was found that there was no significant difference (Table 4).

## 4. Discussion

The results above revealed that stray cats and dogs in the Qinghai–Tibetan Plateau Area, China harbor both *T. gondii* and *N. caninum* infection as confirmed by the antibodies and antigen detection of the indirect ELISA tests and qPCRs. This suggests that stray cats and dogs play key roles in the transmission and prevalence of *T. gondii* and *N. caninum* in the plateau area. To the best of our knowledge, this is the first report of *T. gondii* and *N. caninum* infection in cats and dogs in the QTPA and the first determination of *N. caninum* infection in cats in China.

To detect *T. gondii* and *N. caninum* infection, serological diagnostic methods such as the ELISA and the indirect fluorescent antibody test (IFAT) have been developed [9,14]. Because of the highly specific and sensitive characteristics, ELISAs based on the specific antigens derived from *T. gondii* and *N. caninum* have been used to perform the serological testing for these two parasitic infections in a large number of animal samples [9,14]. Among them, the four specific antigens of *Tg*SAG1, *Tg*GRA1, *Nc*SAG1, and *Nc*GRA7 are the most common [22,23,24,25,26]. The above four-antigen-based ELISA presented its immunodominant nature and could determine specific antibodies with high sensitivity from both acute and chronic *T. gondii* and *N. caninum* infection when used either alone or in combination with the other proteins. Although the SAG1 and GRA7 antigens of *T. gondii* and *N. caninum* were identified expressed in the different stages of parasitic life cycles (tachyzoite or bradyzoite stages), a combination of recombinant-protein-based ELISAs offered the best evidence for the diagnosis of *T. gondii* and *N. caninum* infection. Therefore, this study selected the combination of these two specific antigen-based serological diagnostic methods to detect the specific *Toxoplasma* and *Neospora* IgG and IgM antibodies in animal samples, respectively, which provided the best potential for the diagnosis of the two parasites in stray cats and dogs in the QTPA. Furthermore, the qPCR methods based on *Tg*B1 and *Nc*Nc5 gene amplification from the double-stranded DNA have been widely used for the detection of *T. gondii* and *N. caninum* in species tissues or feces [27,28,29,30,31]. This study also developed molecular diagnostic methods for parasitic detection in stray cat and dog feces, which provided a third indicator of the current determination.

The cat is the most important animal in the epidemiology of toxoplasmosis, because it is the only host that can excrete the environmentally resistant oocysts threatening the intermediate hosts, including humans [32]. A number of studies conducted on toxoplasmosis in the cat population across the globe using diverse serological techniques found that *T. gondii* is distributed globally [21,32,33]. Seroprevalence in cats has been found in many studies in different countries of the world from 2010 to 2020 including those conducted in the USA (19.0%), Brazil (28.6%), Iran (44.6%), Japan (10.9%), and Korea (17.4%) [21,34]. In China, although cats from some provinces were infected by *T. gondii* with infection rates from 2.5% to 60.0% and an overall infection rate of 20.2% in China before 2020, there have been no epidemiological studies on *T. gondii* infections in cats in the district of Qinghai, in the QTPA. The current study provided the first documented evidence on *T. gondii* infection in stray cats in the QTPA and revealed the high occurrence of IgG antibodies at 41.4% and IgM antibodies at 44.5% with the indirect ELISA method in stray cats in Qinghai province. Moreover, one fecal sample of a stray cat showed acute *T. gondii* infection using qPCR-based *Tg*B1. These results indicate that severe acute or chronic *Toxoplasma* infection also exists in cats with a high infection rate in Qinghai province, in the Qinghai–Tibet Plateau Area, which is the most direct evidence of a *Toxoplasma* epidemic in cats in the plateau area.

On the other hand, the epidemiology of *N. caninum* in cats has been barely researched [35]. Additionally, there is no published record of the serology and molecular prevalence of *N. caninum* infection in cats in China. Compared with the previous sero-epidemiological studies from different countries, the frequency of *N. caninum*-seropositive cats in this study was significantly higher [35,36,37,38]. A previous study showed that cats were positive for *N. caninum* related to the presence of *N. caninum* in its live environment and in animals that formed prey for cats [39]. In this regard, the high positivity rates found in the stray cats of the current study indicates that the animals had high exposure to *N. caninum* in the QTPA. Among them, the results of this study showed a high rate of *N. caninum* IgM seropositivity and a low rate of IgG seropositivity in cats, suggesting that most of these stray cats were in the early stage of infection, and only a small number of cats had previously been infected with *N. caninum* and persisted in the body for a long time. This may be because these cats are living in the same environment as dogs infected with *N. caninum*, thus causing recent infections in cats. However, it was not known that laboratory-detectable antibodies to *N. caninum* in cats, even IgM, could represent clinical disease development, and there is still little information on the pathogenesis of neosporosis in cats. Our results update the feline *N. caninum* seropositivity data and also provide the latest data for the study on the epidemiology of neosporosis.

The results of the present study indicated the high seroprevalence of *N. caninum* and *T. gondii* IgG and IgM antibodies in stray dogs in Qinghai province, in the QTPA, compared with in dogs in Jilin, Henan, Heilongjiang, and Anhui provinces, China [40,41,42]. This is the first report of *N. caninum* and *T. gondii* infection in dogs in the Qinghai–Tibet Plateau Area. Oocysts of *N. caninum* shed from dogs are the key factor in the epidemiology of neosporosis in animals; hence, detecting the fecal infection of this parasite has clinical significance [13,15]. Our molecular qPCR assay indicated that six dogs in the QTPA were positive for *N. caninum* parasites in the fecal samples available and the quantitative analysis found high parasite burden. This indicates that the environment may have been contaminated with *N. caninum* parasites. Furthermore, *T. gondii* infection in dogs is of epidemiological and clinical importance. The current study not only found seropositive for *T. gondii* but also calculated a high parasite burden in 16 dog fecal samples using a standard curve prepared by qPCR with the different numbers of *T. gondii*. A previous study determined *T. gondii* DNA was found in dog feces in the USA; dogs can ingest *T. gondii*-infected cat feces, and these oocysts remain viable after passage through the digestive tract of the dog [43]. This study provides key information for dogs or other animals exposed to a contaminated environment with the two parasites, the identification and epidemiology of *N. caninum* and *T. gondii* in dogs should help prevention of neosporosis and toxoplasmosis in the tested area.

Some cat and dog samples currently tested in this study showed both indicators, IgG and IgM, were seropositive in *T. gondii* or *N. caninum* detections and a higher occurrence of both *T. gondii* antibodies at 50.3%, which were the majority of the positive samples as compared to *N. caninum* at 11.4%, while there was a higher *N. caninum* single-IgM seropositive rate (50.9%) than *T. gondii* (3.4%). Generally, the IgG antibodies rise to protective levels after infection and remain detectable for years, while the lower occurrence of IgM antibodies is within days to a couple of weeks; moreover, cat *T. gondii* positive for IgG + IgM are proposed to be chronic reactivated cases [33]. Therefore, the present study suggests the prevalence of acute neosporosis and chronic re-emergence of toxoplasmosis in stray cats and dogs in the testing area, and the fact that the current molecular detection found both parasites in feces may also prove this. Importantly, cases of coinfections with *T. gondii* and *N. caninum* were present in both cat and dog samples tested in this study indicating the importance of dogs and cats in the prevalence of these two similar parasites. However, no significant differences were found between sexes and between the overall infection rates of *T. gondii* and *N. caninum* in this study, which suggests these may not be the key factors for the prevalence of the two parasitic diseases in this region.

## 5. Conclusions

This study is the first documentation of *T. gondii* and *N. caninum* infection, using serological and molecular diagnostic methods in stray cats and dogs in the plateau area, and the determination of *N. caninum* infection in the cats in China. This provides the latest valuable data on the epidemiology study of toxoplasmosis and neosporosis in cats and dogs in China. Although this study found a high positive rate of *T. gondii* and *N. caninum* in dogs and cats in the study area, more attention is needed to prevent the mutual transmission in the two definitive hosts between toxoplasma and neospora. There might be threats of spillover to other species through environmental pollution. However, the current study was limited to either serum or feces collected from one animal, with no way to examine whether the IgM-seropositive samples and the parasite-DNA-positive samples were the same or not. Thus, future studies should assess the prevalence of *T. gondii* and *N. caninum* infections in animals of the definitive and intermediate hosts, including pet and stray cats and dogs and other animals.

## Figures and Tables

**Figure 1 animals-12-01390-f001:**
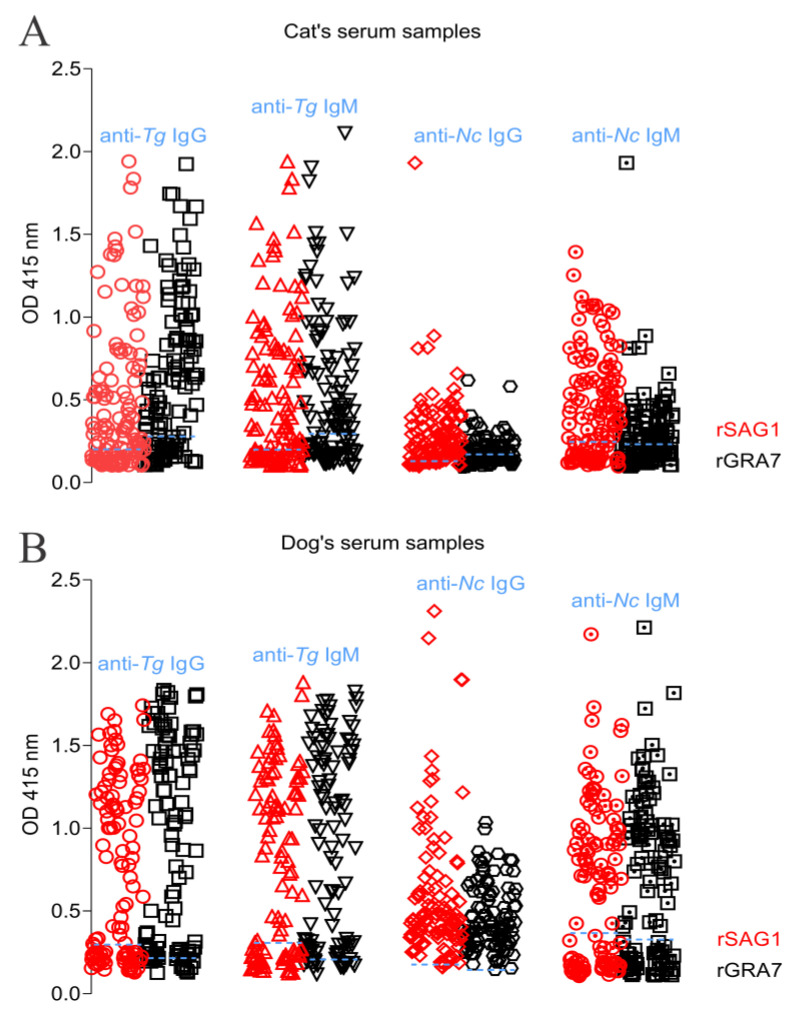
Serological detection of *T. gondii* and *N. caninum* infection in stray cats and dogs in the QTPA. (**A**) The *T. gondii-* and *N. caninum*-specific IgG and IgM antibodies in stray cats using the indirect ELISA based on *rTg*/*Nc*SAG1 and GRA7. (**B**) The *T. gondii* and *N. caninum* specific-IgG and IgM antibodies in stray dogs using the indirect ELISA based on *rTg*/*Nc*SAG1 and GRA7.

**Figure 2 animals-12-01390-f002:**
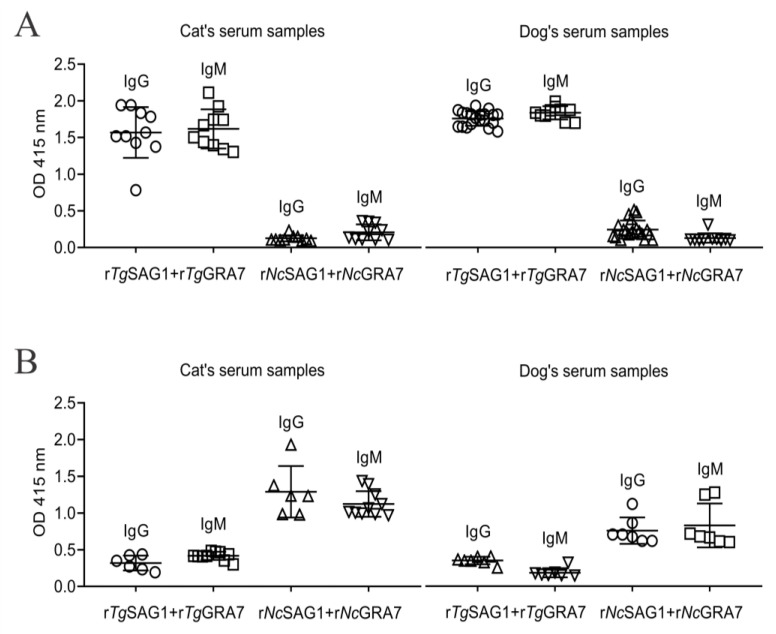
Cross-reaction analysis between the recombinant proteins of *T. gondii* and *N. caninum* and another parasitic antibody. (**A**) Results of the interaction of the recombinant proteins of *T. gondii* and *N. caninum* with *T. gondii*-positive sera of stray dogs and cats, respectively. (**B**) Results of the interaction of the recombinant proteins of *T. gondii* and *N. caninum* with *N. caninum*-positive sera of stray dogs and cats, respectively.

**Figure 3 animals-12-01390-f003:**
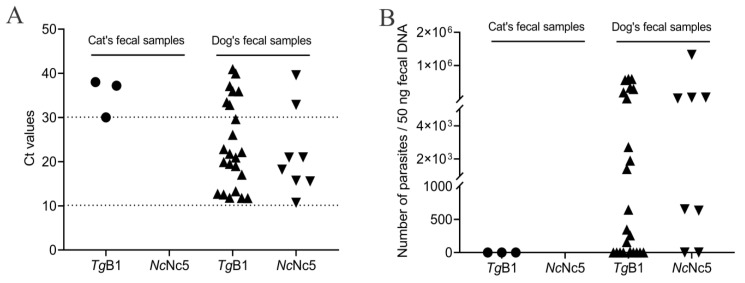
Molecular detection of *T. gondii* and *N. caninum* infection in stray cats and dogs in the QTPA using qPCR based on *Tg*B1 and *Nc*Nc5. (**A**) The Ct value of qPCR; the critical point was 10 and 30. (**B**) The parasite burdens of *T. gondii* and *N. caninum* in 50 ng fecal DNAs using a standard curve prepared by qPCR with two different parasite numbers.

**Table 1 animals-12-01390-t001:** Seroprevalence of *T. gondii-* and *N. caninum*-specific IgG and IgM among stray cats and dogs in the QTPA, China.

Parasite	Animal	No. of Tested	Total IgG-Seropositive		Total IgM-Seropositive		Both IgG and IgM Positive		Single-IgG-Seropositive		Single-IgM-Seropositive	
			Positive	Prevalence % (95% CI)	Positive	Prevalence % (95% CI)	Positive	Prevalence % (95% CI)	Positive	Prevalence % (95% CI)	Positive	Prevalence % (95% CI)
*T. gondii*	Cat	128	53	41.4 (32.9–49.9)	57	44.5 (35.9–53.1)	46	35.9 (27.6–44.2)	7	5.4 (1.5–9.4)	11	8.6 (3.7–13.4)
	Dog	224	134	59.8 (53.4–66.2)	132	58.9 (52.5–65.4)	131	58.5 (52.0–64.9)	3	1.3 (0.2–2.8)	1	0.4 (0.4–1.3)
	Total	352	187	53.1 (47.9–58.3)	189	53.7 (48.5–58.9)	177	50.3 (45.1–55.5)	10	2.8 (1.1–4.6)	12	3.4 (1.5–5.3)
*N. caninum*	Cat	128	2	1.6 (0.6–3.7)	74	57.8 (49.3–66.4)	2	1.6 (0.6–3.7)	0	-	72	56.3 (47.7–64.8)
	Dog	224	40	17.9 (12.8–22.9)	145	64.7 (58.5–71.0)	38	17.0 (12.0–21.9)	2	0.8 (0.3–2.1)	107	47.8 (41.2–54.3)
	Total	352	42	11.9 (8.5–15.3)	219	62.2 (57.2–67.3)	40	11.4 (8.0–14.7)	2	0.5 (0.2–1.4)	179	50.9 (45.6–56.1)

95% CI: 95% Confidence Interval.

**Table 2 animals-12-01390-t002:** Molecular infection rates of *T. gondii* and *N. caninum* among stray cats and dogs in the QTPA, China.

Animal	No. of Tested	*T. gondii* Positive		*N. caninum* Positive		Both *T. gondii* and *N. caninum* Positive	
		Positive	Prevalence % (95% CI)	Positive	Prevalence % (95% CI)	Positive	Prevalence % (95% CI)
Cat	18	1	5.6 (5.0–16.1)	0	-	0	-
Dog	110	16	14.5 (8.0–21.1)	6	5.5 (1.2–9.7)	1	0.9 (0.9–2.7)
Total	128	17	13.3 (7.4–19.2)	6	4.7 (1.0–8.3)	1	0.8 (0.7–2.3)

95% CI: 95% Confidence Interval.

**Table 3 animals-12-01390-t003:** Coinfection with *T. gondii* and *N. caninum* in stray cats and dogs in the QTPA, China.

Animal	No. of Tested	Positive for at Least One Indicator		IgG		IgM		qPCR	
		Positive	Prevalence % (95% CI)	Positive	Prevalence % (95% CI)	Positive	Prevalence % (95% CI)	Positive	Prevalence % (95% CI)
Cat	146	81	55.5 (47.4–63.5)	1	0.7 (0.7–2.0)	52	35.6 (27.8–43.4)	0	-
Dog	334	167	50.0 (44.6–55.4)	37	11.1 (7.7–14.4)	111	33.2 (28.2–38.3)	1	0.3 (0.3–0.9)
Total	480	248	51.7 (47.2–56.1)	38	7.9 (5.5–10.3)	111	23.1 (19.4–26.9)	1	0.2 (0.2–0.6)

95% CI: 95% Confidence Interval.

**Table 4 animals-12-01390-t004:** Toxoplasmosis and neosporosis in stray cats and dogs in the QTPA, China.

Parasitic Diseases	Animal	Positive for at Least one Indicator %	Gender	Diagnostic Test	No. of Tested	No. of Positive	Prevalence %	95% CI	*p*-Value
Toxoplasmosis	Cat	43.8 (64/146)	Male	IgG	55	22	40.0	27.1–52.9	0.8925
				IgM	55	24	43.6	30.5–56.7	
				qPCR	10	1	10.0	8.6–28.6	
			Female	IgG	73	31	42.5	31.1–53.8	
				IgM	73	33	45.2	33.8–56.6	
				qPCR	8	0	0	-	
	Dog	44.9 (150/334)	Male	IgG	105	63	60.0	50.6–69.4	
				IgM	105	62	59.0	49.6–68.5	
				qPCR	51	8	15.7	5.7–25.7	
			Female	IgG	119	71	59.7	50.8–68.5	
				IgM	119	70	58.8	50.0–67.7	
				qPCR	59	8	13.6	4.8–22.3	
Neosporosis	Cat	50.7 (74/146)	Male	IgG	55	0	0	-	0.509
				IgM	55	30	54.5	41.4–67.7	
				qPCR	10	0	0	-	
			Female	IgG	73	2	2.7	1.0–6.5	
				IgM	73	44	60.3	49.0–71.5	
				qPCR	8	0	0	-	
	Dog	45.2 (151/334)	Male	IgG	105	13	12.4	6.1–18.7	
				IgM	105	57	54.3	44.8–63.8	
				qPCR	51	1	2.0	1.8–5.8	
			Female	IgG	119	27	22.7	15.2–30.2	
				IgM	119	88	73.9	66.1–81.8	
				qPCR	59	5	8.5	1.4–15.6	

95% CI: 95% Confidence Interval.

## Data Availability

Data is contained within the article.

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
