# Peer review of "Toxoplasma gondii and Neospora caninum Infections in Stray Cats and Dogs in the Qinghai–Tibetan Plateau Area, China"

_animals, 2022, doi:10.3390/ani12111390_

Round 1
Reviewer 1 Report
The manuscript “Toxoplasma gondii and Neospora caninum infection in stray cats and dogs on Qinghai-Tibetan Plateau Area, China” by Yang and colleagues shows a study on the seroprevalence and molecular epidemiology of Toxoplasma gondii and Neospora caninum using cats and dogs serum and feces samples.
It was not clear in the text if the authors collected samples from a single shelter in Qinghai. Did you perform a statistical study to define the minimum number of animals per shelter and/or the minimum number of shelters that represent the entire QTP area?
Authors name: “Jixu Li1, 2, 3 and *” - “and” should be superscript
Introduction - If the authors want to inform the reader about the QTP area, please give more information. You can include a simple map, showing where are the Qinghai and Tibet provinces
Line 54 - “(an average elevation of more than 4000 m above sea level)” - Please also include the size of the area
Line 55 - “Qinghai province”, in China, “is an important part of the QTPA”
Line 58 and 63 - As we are in the introduction, you can write the full name of the species “Toxoplasma gondii” and “Neospora caninum”
Line 72 - “surface antigen 1” (SAG1) “and dense granule protein 7” (GRA7)
Material and Methods item needs more details.
Line 83 - Sample collection: The authors collected 352 serum samples from cats (n = 128) and dogs (n = 224). What is the total number of animals in the shelter? Did you collect material from all the animals? Please describe it better.
How were the blood samples collected? Once in the lab, how were the sera separated? If by centrifugation, at how many x g and time? Were samples used immediately or stored? Please describe.
How was the feces collection? Was the feces collected fresh? Does the number of samples correspond to different animals? Or were feces from the same animal collected on different days? Please describe better.
Line 88 - I don't understand what you mean by "non-corresponding". You can explain better or remove from the sentence.
Line 98/99 – Please include the AbID code
Line 100 - “ABTS,” please remove comma
Line 110 - DNA isolation and quantitative PCR (qPCR) detection: Please further detail the PCR reaction. qPCR was performed in duplicate or triplicate? Which equipment was used? Please include mix components and concentrations, temperature and reactions time. What was the negative control used?
Line 117 – “The Ct value is between 10-30, we consider the result to be valid” - Did both primers (TgB1 and Nc5) have the same standard curve (between 10 and 30)? Was the efficiency of the primers 100%?
Line 240 – “N. caninum” in italic
Author Response
Response to Reviewer 1 Comments
Thank you Dr. for taking the time to review my manuscript (Manuscript ID: animals-1689724). I revised the manuscript based on your suggestions.
Point 1: It was not clear in the text if the authors collected samples from a single shelter in Qinghai. Did you perform a statistical study to define the minimum number of animals per shelter and/or the minimum number of shelters that represent the entire QTP area?
Response 1: Thank you for your question.
We chose the largest shelter in the Qinghai-Tibet Plateau to collect samples. The stray animals in this shelter came from many regions in Qinghai Province, with the largest number. The current city is the largest city on the Qinghai-Tibet Plateau and the largest population, which can represent the infection of stray animals in the Qinghai-Tibet Plateau.
Point 2: Authors name: “Jixu Li1, 2, 3and *” - “and” should be superscript
Response 2: Thanks for your reminding.
We changed “Jixu Li1, 2, 3and *” to “Jixu Li1, 2, 3*”.(Line 5)
Point 3: Introduction - If the authors want to inform the reader about the QTP area, please give more information. You can include a simple map, showing where are the Qinghai and Tibet provinces.
Response 3: Thanks for your reminding.
We supplemented information about the Qinghai-Tibet Plateau.(Line 56-57)
Point 4: Line 54 - “(an average elevation of more than 4000 m above sea level)” - Please also include the size of the area
Response 4: Thanks for your reminding.
We added the size of the Qinghai-Tibet Plateau area, which is about 2.5 million square kilometers. (Line 56-57)
Point 5: Line 55 - “Qinghai province”, in China, “is an important part of the QTPA”
Response 5: Thanks for your revision.
We revised this sentence. Changed “Qinghai province is an important part of the QTPA” to “Qinghai province in China is an important part of the QTPA”.(Line 57-58)
Point 6: Line 58 and 63 - As we are in the introduction, you can write the full name of the species “Toxoplasma gondii” and “Neospora caninum”
Response 6: Thanks for your reminding.
We added the full name of the species “Toxoplasma gondii” and “Neospora caninum”. (Line 62 and 67)
Point 7: Line 72 - “surface antigen 1” (SAG1) “and dense granule protein 7” (GRA7)
Response 7: Thanks for your revision.
We supplemented the abbreviated forms of “surface antigen 1” and “dense granule protein 7”. (Line 77 and 78)
Point 8: Line 83 - Sample collection: The authors collected 352 serum samples from cats (n = 128) and dogs (n = 224). What is the total number of animals in the shelter? Did you collect material from all the animals? Please describe it better.
Response 8: Thank you for your question.
We described in detail of the information about serum sample collection. There are 352 stray dogs and cats in the shelter, including 224 dogs and 122 cats. There are two types of rearing: free-range and captive. We collected serum samples from all animals. (Line 89 - 93)
Point 9: How were the blood samples collected? Once in the lab, how were the sera separated? If by centrifugation, at how many x g and time? Were samples used immediately or stored? Please describe.
Response 9: Thank you for your question.
We selected the cephalic vein of dogs and cats for blood collection, these blood samples were kept at 37℃ for 30 min, then were centrifuged at 5000 rpm for 10 min, 4℃ to separate and harvest sera and stored at -20℃ until used.
We have made a corresponding description in the manuscript. (Line 96- 98)
Point 10: How was the feces collection? Was the feces collected fresh? Does the number of samples correspond to different animals? Or were feces from the same animal collected on different days? Please describe better.
Response 10: Thank you for your question.
For the feces, we collect are all fresh. There only a part of animal in the shelter has its own cage, so we only collected feces from these animals to rule out the possibility that feces from the same animal could be collected twice.
We have made a corresponding description in the manuscript. (Line 98 - 101)
Point 11: Line 88 - I don't understand what you mean by "non-corresponding". You can explain better or remove from the sentence.
Response 11: Thanks for your reminding.
I'm very sorry for the misunderstanding. Since we only collected feces sample from some animals that have their own cage, we used "mismatched serum and fecal samples". We removed "non-corresponding" from this sentence, and added detailed information about sampling in the next paragraph.(Line 93/96-101)
Point 12: Line 98/99 – Please include the AbID code
Response 12: Thanks for your reminding.
We added the AbID code in the manuscript. (Line 110-112)
Point 13: Line 100 - “ABTS,” please remove comma
Response 13: Thanks for your reminding.
We deleted the comma after “ABTS”.(Line 112)
Point 14: Line 110 - DNA isolation and quantitative PCR (qPCR) detection: Please further detail the PCR reaction. qPCR was performed in duplicate or triplicate? Which equipment was used? Please include mix components and concentrations, temperature and reactions time. What was the negative control used?
Response 14: Thank you for your question.
The amplification was performed with 50 ng DNA, 5 μL PowerUp SYBR Green Master Mix (Thermo Fisher Scientific, Inc., Waltham, MA), and 500 nM concentrations of gene-specific primers in a 10 μL total reaction volume using a standard protocol recommended by the manufacturer (2 min at 50°C, 10 min at 95°C, and then 40 cycles of 95°C for 15 s and 60°C for 1 min). Amplification, data acquisition, and dataanalysis were carried out using the ABI Prism 7900HT sequence detection system (Applied Biosystems), and the cycle threshold values (CT) were calculated. The qPCR was performed in duplicate, and we did not use negative controls. (Line 130-136)
Point 15: Line 117 – “The Ct value is between 10-30, we consider the result to be valid” - Did both primers (TgB1 and Nc5) have the same standard curve (between 10 and 30)? Was the efficiency of the primers 100%?
Response 15: Thank you for your question.
These two primers have been widely used in the detection of T. gondii and N. caninum. The previous studies in our laboratory have identified these two pairs of primers using Lab-grown parasites (results not shown), and they can be effectively used to detection of these two parasites.
Point 16: Line 240 – “N. caninum” in italic
Response 16: Thanks for your reminding.
We changed “N. caninum” to “N. caninum”(in italic).(Line 267)

Reviewer 2 Report
Yang et al. report results on Toxoplasma gondii and Neospora caninum infection in stray cats and dogs on Qinghai-Tibetan Plateau Area, China. Serological and molecular tests were applied for this research.
In terms of qPCR detection of N. caninum in feline feces, dogs are the definitive hosts for this parasite, and its oocysts do not generally shed in the feces of intermediate hosts.
Comments:
Title- please correct as " Toxoplasma gondii and Neospora caninum infections in stray cats and dogs on Qinghai-Tibetan Plateau Area, China" in the title.
Line 5- Delete “and*”.
Line 27, Line 48- Please correct “could be” for "should be".
Line 58, 63- Do not use abbreviation at the beginning of sentence or please consult with editorial team.
Line 84- describe about sampled dogs (stray or owned dogs).
Line 102,106, 236- Why did you examine N. caninum in cat’s feces by qPCR method? It is not surprising that no N. caninum DNA was detected in feces of cat. Dogs are the definitive hosts for this parasite, and its oocysts do not generally shed in the feces of intermediate hosts. N. caninum usually develop as tissue cyst in intermediate hosts.
Line 160 and 180- Font sizes were differences in the two paragraphs.
Line 193- could be
Line 202- Please rephrase the sentence “especially use to detect----“.
Line 239- Please describe how many percent.
Line 240- Italic for N. caninum
Line 284- Infections.
Results- It would be interesting to know whether the IgM seropositive samples and the parasite's DNA positive samples were the same or not.
References- Please correct italic for scientific names.
Author Response
Response to Reviewer 2 Comments
Point 1: Title- please correct as "Toxoplasma gondii and Neospora caninum infections in stray cats and dogs on Qinghai-Tibetan Plateau Area, China" in the title.
Response 1: Thanks for your revision.
We changed “Toxoplasma gondii and Neospora caninum infection in stray cats and dogs on Qinghai-Tibetan Plateau Area, China” to “Toxoplasma gondii and Neospora caninum infections in stray cats and dogs on Qinghai-Tibetan Plateau Area, China”.(Line 2-3)
Point 2: Line 5- Delete “and*”.
Response 2: Thanks for your reminding.
We deleted "and".(Line 5)
Point 3: Line 27, Line 48- Please correct “could be” for "should be".
Response 3: Thanks for your revision.
We corrected “could be” for "play".(Line 27/49)
Point 4: Line 58, 63- Do not use abbreviation at the beginning of sentence or please consult with editorial team.
Response 4: Thanks for your reminding.
We added the full name of the species “Toxoplasma gondii” and “Neospora caninum”. (Line 62 and 67)
Point 5: Line 84- describe about sampled dogs (stray or owned dogs).
Response 5: Thanks for your reminding.
All the animals we sampled were stray animals, which have been described in the article.(Line 89)
Point 6: Line 102,106, 236- Why did you examine N. caninum in cat’s feces by qPCR method? It is not surprising that no N. caninum DNA was detected in feces of cat. Dogs are the definitive hosts for this parasite, and its oocysts do not generally shed in the feces of intermediate hosts. N. caninum usually develop as tissue cyst in intermediate hosts.
Response 6: Thank you for your question.
Although dogs are the definitive host of N. caninum, several reports have detected N. caninum DNA from cat feces, so we also detect in this study.
Such as “Molecular detection and phylogenetic analysis of Neospora caninum in various hosts from Iran” (DOI: 10.1016/j.cimid.2021.101737)
Point 7: Line 160 and 180- Font sizes were differences in the two paragraphs.
Response 7: Thanks for your reminding.
We adjusted to the font size. (Line 201-207)
Point 8: Line 193- could be
Response 8: Thanks for your reminding.
We corrected “could be” for "play".(Line 215)
Point 9: Line 202- Please rephrase the sentence “especially use to detect----“.
Response 9: Thanks for your reminding.
We changed “It is especially the case for the TgSAG1, TgGRA1, NcSAG1, and NcGRA7.” to “Among them, the four specific antigens of TgSAG1, TgGRA1, NcSAG1, and NcGRA7 are the most common.”(Line 224-225)
Point 10: Line 239- Please describe how many percent.
Response 10: Thanks for your reminding.
We changed “while no feces of cats were molecularly positive for N. caninum” to “while the molecular positivity for N. caninum in feces of cat was 0%”(Line 263-264)
Point 11: Line 240- Italic for N. caninum
Response 11: Thanks for your reminding.
We changed “N. caninum” to “N. caninum” (in italic). (Line 267)
Point 12: Line 284- Infections.
Response 12: Thanks for your revision.
We changed “infection” to “infections”. (Line 306)
Point 13: Results- It would be interesting to know whether the IgM seropositive samples and the parasite's DNA positive samples were the same or not.
Response 13: Thanks for your reminding.
Only some of the feces and serum of the animals we collected corresponded to each other, so unfortunately we were unable to do this kind of analysis.
Point 14: References- Please correct italic for scientific names.
Response 14: Thanks for your reminding.
We corrected italic for scientific names.

Reviewer 3 Report
Reviewer comments for manuscript ID animals-1689724 entitled Toxoplasma gondii and Neospora caninum infection in stray cats and dogs on Qinghai-Tibetan Plateau Area, China’
General comments
The authors have worked on a very important issue of public health importance that has global implications. Toxoplasmosis and Neosporidiosis are re-emerging diseases having zoonotic potential and epidemiological studies are needed in different geographical areas for revealing the spread of the infection. Such studies are the benchmarks on which the future public health interventions will be based. I congratulate the authors for this nice work.
I suggest the authors to please detail the public health importance of these two species worldwide in the introduction section to highlight the importance of this work. Discussion section is weak and has repetitions. I would like to see a more indepth analysis of the results.
It is an interesting work and I would like to see more work in the write up to justify this remarkable research.
Specific comments
Line 15: Please replace ‘husbandries’ with ‘husbandry’
Line 27: Please replace ‘should be’ with ‘play’
Lines 33-37: It is a huge sentence. Please reduce it into multiple sentences.
Lines 43-46: Please rewrite into smaller sentences.
Lines 46-47: Please reframe ‘Further analysis showed that no significant differences were found genders’ as ‘Further analysis showed that no significant gender differences were found’
Line 48: Please replace ‘should be’ with ‘play’
Line 56: Please replace ‘husbandries’ with ‘husbandry’
Line 61: Please replace ‘lived’ with ‘living’
Line 87: Please replace ‘animals’ with ‘species’
Line 106: Please replace ‘judged’ with ‘considered’
Line180: Please delete ‘lightly’
Line 191: Please replace ‘present’ with ‘harbour’ and replace ‘by the antibodies’ with ‘as confirmed by the antibodies’
Line 193: Please replace ‘should be’ with ‘play’
Line 215: Please replace ‘animal’ with ‘species’
Lines 217-19: Please complete the sentence.
Line 225: Please reframe ‘firstly recorded’ as ‘the first documented evidence on’
Line226: Please replace ‘documented’ with ‘revealed’
Lines 232-34: Please reframe as ‘On the other hand, the importance and epidemiology of N. caninum in cats have been barely researched [36]. There is no published record of the serology and molecular prevalence of N. caninum infection in cats in China till date’
Lines 240-42: Please reframe this sentence, it lacks clarity.
Line 271: Please delete ‘considered’
Line 284: Please replace ‘firstly demonstrates’ with ‘is the first documentation of ‘
Lines 288-91: Please reframe as ‘Although this study found a high positive rate of T. gondii and N. caninum in dogs and cats in the study area, more attention is needed to prevent the mutual transmission in two definitive hosts between Toxoplasma and Neospora. These might be threats of spill over to other species through environmental pollution’
Lines 291-93: I am sorry I am not able to understand this line. Please clarify.
Author Response
Response to Reviewer 3 Comments
Point 1: Line 15: Please replace ‘husbandries’ with ‘husbandry’
Response 1: Thanks for your revision.
We changed “husbandries” to “husbandry” (Line 15)
Point 2: Line 27: Please replace ‘should be’ with ‘play’
Response 2: Thanks for your revision.
We changed “should be” to “play” (Line 27)
Point 3: Lines 33-37: It is a huge sentence. Please reduce it into multiple sentences.
Response 3: Thanks for your reminding.
We reduced this sentence into multiple sentences. “Therefore, the aim of this study was to perform the indirect ELISA tests based on recombinant TgSAG1, TgGRA1, NcSAG1 and NcGRA7 proteins to establish a detailed record of the seroprevalence of T. gondii and N. caninum-specific IgG and IgM antibodies in serum samples, and to develop the qPCR amplification based on TgB1 and NcNc5 genes to do the molecular epidemiology in feces from the stray cats and dogs on the QTPA.” (Line 34-38)
Point 4: Lines 43-46: Please rewrite into smaller sentences.
Response 4: Thanks for your reminding.
We revised this sentence. “On the other hand, 1 of 18 cat fecal samples was successfully amplified within 10 to 30 of Ct value while no cat was positive for neosporosis. Moreover, a higher prevalence of toxoplasmosis in stray dogs (14.5%, 16/110) than of neosporosis (5.5%, 6/110) with the different parasite numbers were found.” (Line 44-47)
Point 5: Lines 46-47: Please reframe ‘Further analysis showed that no significant differences were found genders’ as ‘Further analysis showed that no significant gender differences were found’
Response 5: Thanks for your revision.
We changed “Further analysis showed that no significant differences were found genders” to “Further analysis showed that no significant gender differences were found” (Line 47-48)
Point 6: Line 48: Please replace ‘should be’ with ‘play’
Response 6: Thanks for your revision.
We changed “should be” to “play” (Line 49)
Point 7: Line 56: Please replace ‘husbandries’ with ‘husbandry’
Response 7: Thanks for your revision.
We changed “husbandries” to “husbandry” (Line 61)
Point 8: Line 61: Please replace ‘lived’ with ‘living’
Response 8: Thanks for your revision.
We changed “lived” to “living” (Line 66)
Point 9: Line 87: Please replace ‘animals’ with ‘species’
Response 9: Thanks for your revision.
We changed “animals” to “species” (Line 92)
Point 10: Line 106: Please replace ‘judged’ with ‘considered’
Response 10: Thanks for your revision.
We changed “judged” to “considered” (Line 119)
Point 11: Line180: Please delete ‘lightly’
Response 11: Thanks for your revision.
We deleted ‘lightly’ (Line 201)
Point 12: Line 191: Please replace ‘present’ with ‘harbour’ and replace ‘by the antibodies’ with ‘as confirmed by the antibodies’
Response 12: Thanks for your revision.
We changed “present” to “harbour” and changed “by the antibodies” to “as confirmed by the antibodies” (Line 213-214)
Point 13: Line 193: Please replace ‘should be’ with ‘play’
Response 13: Thanks for your revision.
We changed “should be” to “play” (Line 215)
Point 14: Line 215: Please replace ‘animal’ with ‘species’
Response 14: Thanks for your revision.
We changed “animals” to “species” (Line 237)
Point 15: Lines 217-19: Please complete the sentence.
Response 15: Thanks for your reminding.
We completed the sentence. “A number of studies conducted on toxoplasmosis in the cat population across the globe using diverse serological techniques found that T. gondii is distributed globally.” (Line 243-245)
Point 16: Line 225: Please reframe ‘firstly recorded’ as ‘the first documented evidence on’
Response 16: Thanks for your revision.
We changed “firstly recorded” to “the first documented evidence on” (Line 251-252)
Point 17: Line226: Please replace ‘documented’ with ‘revealed’
Response 17: Thanks for your revision.
We changed “documented” to “revealed” (Line 252)
Point 18: Lines 232-34: Please reframe as ‘On the other hand, the importance and epidemiology of N. caninum in cats have been barely researched [36]. There is no published record of the serology and molecular prevalence of N. caninum infection in cats in China till date
Response 18: Thanks for your revision.
We changed “On the other hand, the importance of N. caninum in cats and cat's impact on the epidemiology of it have been little described [36], and no publish is recorded about the serology and molecular prevalence of N. caninum infection in cats in China to data.” to “On the other hand, the importance and epidemiology of N. caninum in cats have been barely researched [36]. There is no published record of the serology and molecular prevalence of N. caninum infection in cats in China till date.” (Line 259-261)
Point 19: Lines 240-42: Please reframe this sentence, it lacks clarity.
Response 19: Thanks for your reminding.
We reframed this sentence. “A previous study showed that cats were positive for N. caninum related to the presence of N. caninum in its live environment and in animals that formed prey for cats.” (Line 266-268)
Point 20: Line 271: Please delete ‘considered’
Response 20: Thanks for your reminding.
We deleted ‘considered’ (Line 299)
Point 21: Line 284: Please replace ‘firstly demonstrates’ with ‘is the first documentation of ‘
Response 21: Thanks for your revision.
We changed “firstly demonstrates” to “is the first documentation of” (Line 313)
Point 22: Lines 288-91: Please reframe as ‘Although this study found a high positive rate of T. gondii and N. caninum in dogs and cats in the study area, more attention is needed to prevent the mutual transmission in two definitive hosts between Toxoplasma and Neospora. These might be threats of spill over to other species through environmental pollution’
Response 22: Thanks for your revision.
We changed “Although this study found a high positive rate of T. gondii and N. caninum in dogs and cats in this area in this study, we should probably pay more attention to the mutual transmission of two parasites in two definitive hosts between Toxoplasma and Neospora, of the two diseases, as well as the environmental pollution threatens to other animals.” to “Although this study found a high positive rate of T. gondii and N. caninum in dogs and cats in the study area, more attention is needed to prevent the mutual transmission in two definitive hosts between Toxoplasma and Neospora. These might be threats of spill over to other species through environmental pollution” (Line 317-320)
Point 23: Lines 291-93: I am sorry I am not able to understand this line. Please clarify.
Response 23: Thanks for your reminding.
I'm very sorry for the misunderstanding. Since we collected only one type of sample from the same animal, feces or serum, we used "mismatched serum and fecal samples". We modified this sentence to "However, the current study is limited to either serum or feces in one collected animal, and no way to understand whether the IgM seropositive samples and the parasite's DNA positive samples were the same or not. ". (Line321-323)
